# Exosomes Derived from Adipose Tissue-Derived Mesenchymal Stromal Cells Prevent Medication-Related Osteonecrosis of the Jaw through IL-1RA

**DOI:** 10.3390/ijms24108694

**Published:** 2023-05-12

**Authors:** Yi Zheng, Xian Dong, Xinyu Wang, Jie Wang, Shuo Chen, Yang He, Jingang An, Linhai He, Yi Zhang

**Affiliations:** 1Department of Oral and Maxillofacial Surgery, Peking University School and Hospital of Stomatology, Beijing 100081, China; 2First Clinical Division, Peking University School and Hospital of Stomatology, Beijing 100081, China

**Keywords:** medication-related osteonecrosis of the jaw, zoledronate, adipose tissue-derived mesenchymal stromal cells, exosomes, interleukin-1 receptor antagonist, human gingival fibroblasts, primary wound healing

## Abstract

Medication-related osteonecrosis of the jaw (MRONJ) is a severe disease with unclear pathogenesis. Adipose tissue-derived mesenchymal stromal cells (MSC(AT)s) serve as a special source for cell therapy. Herein, we explored whether exosomes (Exo) derived from MSC(AT)s promote primary gingival wound healing and prevent MRONJ. An MRONJ mice model was constructed using zoledronate (Zol) administration and tooth extraction. Exosomes were collected from the conditioned medium (CM) of MSC(AT)s (MSC(AT)s-Exo) and locally administered into the tooth sockets. Interleukin-1 receptor antagonist (IL-1RA)-siRNA was used to knock down the expression of IL-1RA in MSC(AT)s-Exo. Clinical observations, micro-computed tomography (microCT), and histological analysis were used to evaluate the therapeutic effects in vivo. In addition, the effect of exosomes on the biological behavior of human gingival fibroblasts (HGFs) was evaluated in vitro. MSC(AT)s-Exo accelerated primary gingival wound healing and bone regeneration in tooth sockets and prevented MRONJ. Moreover, MSC(AT)s-Exo increased IL-1RA expression and decreased interleukin-1 beta (IL-1β) and tumor necrosis factor-α (TNF-α) expression in the gingival tissue. The sequent rescue assay showed that the effects of preventing MRONJ in vivo and improving the migration and collagen synthesis abilities of zoledronate-affected HGFs in vitro were partially impaired in the IL-1RA-deficient exosome group. Our results indicated that MSC(AT)s-Exo might prevent the onset of MRONJ via an IL-1RA-mediated anti-inflammatory effect in the gingiva wound and improve the migration and collagen synthesis abilities of HGFs.

## 1. Introduction

Medication-related osteonecrosis of the jaw (MRONJ) is a serious disease caused by anti-resorption and anti-angiogenetic drugs used to prevent skeletal events associated with osteoporosis and bone metastases [1]. MRONJ has been associated with severe pain, halitosis, dysphagia, and difficulty chewing, which can seriously affect the quality of life. Thus far, several treatment strategies for MRONJ have been recommended according to different clinical MRONJ stages, yet their therapeutic efficiency remains limited. Additionally, the pathogenesis of MRONJ needs to be further elucidated [2,3].

Bisphosphonate-induced impairment of soft tissue healing is considered a potential mechanism leading to MRONJ [4]. Previous studies suggested gingival healing impairment in MRONJ lesions [5,6]. In addition, promoting primary mucosal closure after tooth extraction could effectively reduce the onset of MRONJ [6,7,8]. Thus, well-healed mucosa might be crucial for the prevention of MRONJ.

Recent studies have shown that stem cell-based therapy prevents MRONJ [9,10,11,12]. Our serial studies have shown that local administration of MSC(AT)s or MSC(AT)-CM can effectively prevent MRONJ by promoting gingival healing and rescuing bone remodeling [6,13]. Moreover, Nifosi et al. proposed using mesenchymal stem cells to treat MRONJ [14]. However, the application of stem cell therapy in clinical treatment is still very limited due to many ethical issues.

Numerous studies have shown that exosomes are important in stem cell translation therapy [15,16]. Exosomes are vesicles ranging from 30 to 130 nm in size. They contain numerous proteins, lipids, and nucleic acid, secreted by cells and delivered to adjacent or distant cells, regulating the target cell’s gene expression, signaling, and function [17,18]. Previous studies found that exosomes derived from stem cells may promote wound healing by promoting the proliferation and migration of skin cells and activating β-catenin signaling [19]. They can also modulate immune responses and inflammation [20] and promote angiogenesis [21]. However, whether MSC(AT)-derived exosomes can promote gingival wound healing and prevent MRONJ has not yet been reported.

In the present study, we explored the effects and mechanisms of MSC(AT)-derived exosomes in promoting gingival wound healing and preventing MRONJ by using an MRONJ-like mouse model and clinical samples.

## 2. Results

### 2.1. Characterization of Exosomes and Hydrogel Loaded with Exosomes

Ultracentrifugation was used to isolate exosomes from MSC(AT)s, as previously reported [22]. Nanoparticle tracking analysis (NTA) analysis showed that the exosomes had a mean size of 103.7 nm and an SD of 29.4 nm (Figure 1A). Transmission electron microscope (TEM) analysis showed spherical morphology (Figure 1B). Furthermore, Western blotting revealed the expression of CD9 and CD63 proteins (exosomal markers) and no expression of calnexin (cell surface markers) (Figure 1C). Confocal scanning showed that PKH26-labelled exosomes (red dots) were uniformly distributed in the hydrogel (Figure 1D). Quantification of the isolated exosomes showed that exosomes were slowly released from the hydrogel over time by BCA assay (Figure 1E).

### 2.2. Exosomes Promote Gingival Wound Closure and Facilitate Socket Bone Regeneration in MRONJ-like Mice

Phosphate buffer saline (PBS), hydrogel, or hydrogel loaded with exosomes were locally administrated into the tooth sockets after tooth extraction (Figure 2A). Two weeks post-extraction, stereoscope observations showed incomplete gingival healing with exposed socket bone in the Zol and the Hydrogel groups, while complete mucosa coverage was observed in the Ctrl and the Exo groups (Figure 2B). MicroCT analysis further showed reduced new bone formation in the extraction socket in the Zol and the Hydrogel groups compared to Ctrl and the Exo groups (Figure 2C). Furthermore, the quantified microCT results showed that bone volume/total volume (BV/TV), bone mineral density (BMD), and trabecular thickness (Tb.Th) were decreased in the Zol group and the Hydrogel groups compared with the Ctrl group (*p* < 0.05, Figure 2D), yet these parameters increased after the local administration of exosomes in the extraction socket (*p* < 0.05, Figure 2D). 

Histologic analysis results showed deficient consecutive epithelial lining, insufficient bone regeneration, and even necrotic bone with empty lacunae in the Zol and the Hydrogel groups, while intact epithelial and connective tissue coverage and new bone formation were found in the Ctrl and Exo groups (Figure 2E and Appendix A). 

Masson staining showed that exosomes promoted greater collagen fiber formation and deposition in the tooth sockets in the Exo group compared to the Zol group and the Hydrogel group (Figure 2F and Appendix A). 

Tartrate-resistant acid phosphatase (TRAP) staining showed a decreased average number of osteoclasts per bone marrow area (#/mm^2^) in the Zol and the Hydrogel groups but also an increased number in the Exo group (all *p* < 0.05, Appendix A). Immunochemistry (IHC) staining for osteocalcin (OCN) revealed increased expression of OCN in the Exo group compared to the Zol group and the Hydrogel group (all *p* < 0.05, Appendix A).

### 2.3. Exosomes Increase IL-1RA Expression and Decrease IL-1β and TNF-α Expression in the Gingival Tissue

In their study, Kou et al. suggested that IL-1RA secretion in gingival mesenchymal stem cells (GMSCs) accelerates wound healing [23]. To determine the underlying mechanism of exosome-mediated wound healing, we performed Immunofluorescent (IF) staining and found that the expression of IL-1RA in gingival tissue increased in the Exo group compared to the Zol group and the Hydrogel group (Figure 3A). Meanwhile, IL-1β and TNF-α expression increased in the gingiva tissue in the Zol group and the Hydrogel group, while Exo reduced these pro-inflammation cytokines’ expression in the Exo group (Figure 3C,E). To further clarify the resource of pro-inflammatory cytokines, we detected the M1/M2 phenotype of macrophages, which could produce a large number of cytokines in the process of wound healing [24,25,26]. Because of the share markers of M1 or M2 macrophages, we characterized the CD86^+^ cells as M1 macrophages and CD206^+^ cells as M2 macrophages. IHC staining results showed large numbers of CD86^+^ M1 macrophages and a few numbers of CD206^+^ M2 macrophages in the gingiva tissues in the Zol group and the Hydrogel group, while Exo can promote macrophage M2 polarization and inhibit macrophage M1 polarization in the Exo group (Appendix A). Furthermore, there was no obvious difference in IL-1RA expression in gingival samples extracted from healthy and MRONJ patients (Figure 3B). Yet, MRONJ patients’ gingival samples lesions had higher levels of IL-1β and TNF-α expression than the healthy controls (Figure 3D,F). 

### 2.4. IL-1RA Deficiency Affects Exosomes in Promoting Gingival Wound Healing in MRONJ-like Mice

Our previous study showed that MSC(AT)s-Exo increases IL-1RA expression and prevents MRONJ. To further explore the functions of IL-1RA in MSC(AT)s-Exo promoting the gingival wound healing process, IF staining was performed, and it revealed that MSC(AT)s contain IL-1RA (Figure 4A), which was further confirmed by Western blotting (Figure 4B). Then, *IL-1RA*-siRNA was used to knock down the expression of IL-1RA in MSC(AT)s and exosomes. The silencing efficiency of *IL-1RA* was demonstrated by real-time PCR and Western blotting (*p* < 0.05, Figure 4C,D). Importantly, *IL-1RA*-siRNA decreased IL-1RA in exosomes (Figure 4E). 

In vivo results showed that IL-1RA-deficient exosomes decrease IL-1RA levels in gingival tissues and partially impair the capacity of MSC(AT)s-Exo in promoting gingival wound healing. Clinical observations showed incomplete gingival coverage was seen in the Exo (si-IL-1RA) group (Figure 5A). MicroCT results showed reduced bone regeneration, BMD, and Tb.Th in the Exo (si-IL-1RA) group (all *p* < 0.05, Figure 5B,C). H&E staining showed incomplete gingival coverage and insufficient bone formation in the tooth extraction of mice in the Exo (si-IL-1RA) group compared with the Exo (si-scramble) group (Figure 5D). IHC staining and IF staining showed the IL-1β and TNF-α expression increased and the IL-1RA expression decreased in the gingival tissues in the Exo (si-IL-1RA) group (Figure 5E–G). Furthermore, IHC staining showed increased numbers of M1 macrophages and decreased numbers of M2 macrophages in the gingiva tissues in the Exo (si-IL-1RA) group compared with the Exo group (Appendix A). These results indicated that IL-1RA carried by exosomes might have a vital role in the exosome-mediated promotion of gingival wound healing and the prevention of the occurrence of MRONJ.

### 2.5. IL-1RA Deficiency in Exosomes Partially Inhibits the Migration and Collagen Synthesis of Gingival Fibroblasts

Fibroblasts are critical in wound healing and tissue regeneration [27]. To further explore the mechanisms of IL-1RA derived from exosomes in promoting gingival healing, in vitro experiments were conducted. The results showed that PKH26-labeled exosomes could be engaged by HGFs at 6 h and 24 h, and the content of phagocytosed exosomes increased over time (Figure 6A). The migration assay further suggested that 10 μM Zol could suppress the migration of HGFs at 24 h and 48 h. Exo (si-scramble) accelerated HGF migration at 48 h, while the migration ability was partially abolished in the Exo (si-IL-1RA) group (all *p* < 0.05, Figure 6C,D). Furthermore, IF staining showed Zol significantly suppressed collagen I (COL1A1) and fibronectin (FN) expression in HGFs, and Exo (si-scramble) effectively promoted COL1A1 and FN expression in Zol-treated HGFs compared with the Exo (siRNA-IL-1RA) group (all *p* < 0.05, Figure 6E–H).

## 3. Discussion

It has been reported that bisphosphonate toxicity to soft tissues may contribute to impaired wound healing, and delayed gingival wound healing facilitates the development of osteonecrosis of the jaw [4]. Previous studies have found that MSC(AT)s or MSC(AT)s-Exo could promote soft tissue healing in various complex conditions. Yu et al. suggested that MSC(AT)s could mitigate severe radiation-induced skin injury [28]. Moreover, MSC(AT)s-Exo was also reported to accelerate cutaneous wound healing in a diabetic wound by promoting vascularization [29] or by inducing miR-128-3p/SIRT1-mediated autophagy [30]. In oral lesions, exosomes promote craniofacial soft tissue regeneration by enhancing Cdc42-mediated vascularization [31] and improve tooth extraction socket healing by promoting angiogenesis [32]. Consistently, our previous studies found that MSC(AT)s and MSC(AT) cell supernatant had the potential to prevent MRONJ [6,13]. Furthermore, other studies have demonstrated that the systematic application of exosomes has an important role in the wound healing of MRONJ lesions [32,33]. To further prevent the accumulation of exosomes in other organs, such as the liver, lung, etc., we locally injected MSC(AT)s-Exo into the extracted tooth socket. 

There are many factors that can affect the wound-healing process [34]. For example, fibroblasts have a critical role in supporting normal wound healing, such as breaking down the fibrin clot, secreting various cytokines, contracting the wound, and supporting the other cells related to effective wound healing [35]. In addition, a dysregulated or chronic inflammatory response may impair wound healing by interfering with normal tissue function and architecture [36]. Macrophages, as essential inflammatory cells, have a variety of functions and can influence all stages of wound healing [26]. In particular, the pro-inflammatory M1 macrophage phenotype could produce numerous mediators and cytokines, such as IL-1β and TNF- α, which may affect wound healing [24,25,26]. Our study found that MSC(AT)s-Exo might effectively accelerate gingival wound healing by improving HGFs’ migration ability and collagen formation abilities. Moreover, our results suggested MSC(AT)s-Exo might reduce the expression of pro-inflammatory cytokines, such as IL-1β and TNF-α, and increase IL-1RA in the gingival tissue of the tooth sockets. IHC staining showed MSC(AT)s-Exo might promote macrophage M2 polarization and inhibit macrophage M1 polarization in the gingival tissue during the wound healing process. Furthermore, we also found that MSC(AT)s-Exo increased the number of osteoclasts and OCN expressions, which suggested the bone remodeling process in MRONJ lesions might be rescued. 

Proteins, lipids, DNA, mRNA, microRNA (miRNA), and noncoding RNA found in exosomes are carried to target cells, having an important role in various physiological and pathological conditions [18]. In this study, we found that MSC(AT)s-Exo was rich in IL-1RA, which might be essential in exosome-meditated gingival wound inflammation suppression and the restoration of cell viability fibroblasts. IL-1RA can effectively block the IL-1β–driven inflammatory signals and modulate immune statuses, such as attenuating the antigen-presenting properties of dendritic cells, inducing an anti-inflammatory phenotype in macrophages, and inhibiting T helper 17 cell differentiation, which provides an immunosuppressive environment and alleviates excessive immune response, thus contributing to tissue repair and regeneration [37,38,39,40,41]. Previous studies showed that IL-1RA promotes chronic diabetes wound healing by re-establishing a pro-healing microenvironment characterized by fewer pro-inflammatory cells, cytokines, senescent fibroblasts, anti-inflammatory cytokines, and growth factors [42]. In addition, IL-1RA deficiency can delay wound healing in a gingival wound model. Kou et al. found that GMSC-derived exosomes containing IL-1RA, or anakinra, a form of recombinant IL-1RA, significantly accelerate gingival wound healing [23]. Consistently, our study also showed that MSC(AT)-derived exosomes elevated IL-1RA expression and reduced the pro-inflammatory cytokines IL-1β and TNF-α in MRONJ-gingival connective tissue, which may be due to the MSC(AT)s-Exo-mediated shift from the M1 macrophage to M2 macrophage phenotype. At the same time, IL-1RA-deficient exosomes aggravated pro-inflammation levels and impaired gingival healing of MRONJ. These results suggested that IL-1RA might have a key function in MSC(AT)-derived exosomes promoting the gingival healing process. 

The present study has a few limitations. First, the detailed mechanism of how IL-1RA derived from MSC(AT)s-Exo prevents MRONJ was not investigated, and more research on macrophages is required. Second, previous studies showed that MSC(AT)s-Exo promoted different effects on various kinds of cells, such as promoting the angiogenesis of human umbilical vein endothelial cells or promoting the migration, proliferation, and osteogenic differentiation of BMSCs [32,43]. Therefore, we supposed that MSC(AT)s-Exo partly prevented MRONJ by regulating BMSCs, osteoblasts, osteoclasts, or vascular endothelial cells, which should be further investigated. 

## 4. Materials and Methods

### 4.1. Preparation and Identification of Exosomes

MSC(AT)s were collected from the cell bank, as described in our previous study [6]. They were cultured in α-modified Eagle medium (α-MEM, C12571500BT, Gibco, Waltham, MA, USA) supplemented with 10% fetal bovine serum (FBS) (10100147, Gibco, Waltham, MA, USA) and 1% penicillin−streptomycin (15140-122, Gibco, Waltham, MA, USA) in a humidified atmosphere containing 5%CO_2_/95% air at 37 °C. After cells reached ~80% confluency, the culture medium was replaced with α-modified Eagle’s medium containing exosome-depleted FBS (EXO-FBS-50A-1, SYSTEM BIOSCIENCES, Palo Alto, CA, USA) and 1% penicillin−streptomycin. Cells were then cultured for an additional 48 h. Consequently, MSC(AT)-CM was collected.

Exosomes were isolated using the following steps: (1) the CM was centrifuged sequentially at 300× *g* for 10 min, 2000× *g* for 30 min, and then at 10,000× *g* for 30 min to remove cells and cell debris; (2) CM supernatants were filtered through a 0.22-mm filter to remove any larger particles; (3) supernatants were ultracentrifuged at 100,000× *g* for 70 min to pellet the EVs. Next, a second washing step was performed by resuspending the EV pellet in 15 mL of PBS (ZLI-9062, ZSGB-BIO, Beijing, China) and carrying out ultracentrifugation at 100,000× *g* for another 70 min; (4) the pellet was then resuspended in PBS. 

NTA (ZetaVIEW S/N 17-310, PARTICLE METRIX, Inning am Ammersee, Germany) was performed with a nanoparticle analyzer (ZetaVIEW 8.04.02, Germany) to measure the distribution and size of isolated exosomes. The shape of exosomes was analyzed using a TEM (Tecnai G2 Spirit BioTwin, FEI, Hillsboro, OR, USA). Antibodies against calnexin (ab133615, Abcam, Cambridge, UK), CD9 (ab263019; Abcam), and CD63 (ab134045, Abcam, Cambridge, UK) proteins were used to detect the surface markers of exosomes by Western blotting. 

### 4.2. Isolation and Culture of Human Gingival Fibroblasts (HGFs)

HGFs were obtained from healthy donors. All subjects signed a written informed consent (PKUSSIRB-202170184). After being thoroughly rinsed with sterile PBS supplemented with antibiotics (2% penicillin−streptomycin) three times, tissues were cut into small pieces with a sterile lancet. Samples were then digested at 37 °C for 30 min with a solution containing dispase II (4 mg/mL, 494207800, Roche, Indianapolis, IN, USA) and collagenase I (2 mg/mL, C0130-1G, Sigma-Aldrich, St. Louis, MO, USA). After digestion, the suspension was centrifuged (1200 rpm, 5 min, resuspended, and filtered using a 70-μm filter). The flow-through and lysed tissues were collected and seeded in Dulbecco’s modified Eagle’s medium (DMEM, C11995500BT, Gibco, Waltham, MA, USA) supplemented with 10% FBS (10100147, Gibco, Waltham, MA, USA) and 1% penicillin−streptomycin in a humidified atmosphere containing 5%CO_2_/95% air at 37 °C. HGFs between the 3rd and 8th passages were used in the experiments.

### 4.3. Exosomes Uptake Assay

Exosomes were labeled with the red fluorescent cell linker PKH-26 (MINI26, Sigma-Aldrich, St. Louis, MO, USA) according to the manufacturer’s protocol to determine their uptake by HGFs. Exosomes diluted in 1 mL Diluent C and 4 μL PKH26 dye diluted in 1 mL Diluent C were incubated together. After 4 min, 2 mL 0.5% bovine serum albumin was added to bind excess dye, after which the labeled exosomes were washed in PBS at 100,000× *g* for 1 h. Next, the exosome pellet was incubated with HGFs for 6 h and 24 h. After each time point, cells were washed twice with PBS and fixed in 4% paraformaldehyde for 15 min. DAPI (ZLI-9557, ZSGB-BIO, China) solution was used to stain nuclei. Images were captured with a confocal imaging system (TCS-SP8, Leica, Wetzlar, Germany).

### 4.4. siRNA Transfection 

Small interfering RNAs (siRNA) were used to silence *IL-1RA* expression in MSC(AT)s. MSC(AT)s (passages 3–5 times) were plated in six-well plates (150,000 cells/well) in a complete medium. The transfection medium was prepared with opti-MEM (Invitrogen, Carlsbad, CA, USA) combined with *IL-1RA*-specific or a nonspecific control siRNA (scramble siRNA) (RiboBio Co., Ltd., Guangzhou, Guangdong, China) and Lipofectamine RNAi-max (Invitrogen, Carlsbad, CA, USA). Then, transfection of *IL-1RA*-specific or nonspecific control siRNA was performed using the Lipofectamine RNAiMAX transfection reagent according to the instructions. After 48 h, MSC(AT)s were homogenized in lysis buffer for RNA and protein extraction, which were subsequently used for quantitative real-time polymerase chain reaction (qRT-PCR) and Western blot experiments. Furthermore, cell supernatants were collected to extract exosomes, as described before. The exosomes derived from IL-1RA-knockdown MSC(AT)s were named Exo (si-IL-1RA), and the control exosomes were named Exo (si-scramble). The double-stranded sequence of siRNA is listed in Appendix A.

### 4.5. Cell Viability

The effect of Zol on HGFs was quantitatively analyzed with the cell counting kit-8 (CCK8) assay (Dojindo, Kumamoto, Japan). Cells were seeded in 96-well plates (2 × 10^3^ cells/100 μL/well) and incubated overnight. Then, the cells were incubated with a gradual concentration of Zol (0, 1, 5, 10, 25, and 50 μM) for 24 h and then with CCK-8 solution (10 µL per well) at 37 °C for 2h in the dark. The absorbance was measured at 450 nm absorbance using a microplate reader (ELX808, Bio Tek, Winooski, VT, USA).

### 4.6. Cell Migration Assay

A scratch wound healing assay was used to evaluate the effect of Zol or/and exosomes on HGF migration. HGFs were seeded in six-well plates and divided into 4 groups: (1) Ctrl group (no Zol treatment); (2) Zol group (cells were incubated with 10 μM Zol for 24 h); (3) Exo (si-scramble) group (cells were incubated with 10 μM Zol for 24 h and administered with 50 μg/mL Exo (si-scramble)); (4) Exo (si-IL-1RA) group (cells were incubated with 10 μM Zol for 24 h and administered with 50 μg/mL Exo (si-IL-1RA)). A line was drawn using a marker on the bottom of the dish. After incubation with 10 μM Zol for 24 h, a sterile 20-μL pipet tip was used to scratch three separate wounds, moving perpendicular to the line. After washing the cells with PBS, 50 μg/mL Exo (si-scramble) or Exo (si-IL-1RA) were added to the culture medium of the Exo (si-scramble) group or the Exo (si-IL-1RA) group. Images of the scratches were taken using an inverted microscope (Olympus, Lake Success, NY, USA) at ×10 magnification at 0, 24, and 48 h after scratching. The percentage of wound closure was calculated using Image J 1.51j8 software.

### 4.7. Cell Immunofluorescent Staining 

MSC(AT)s were seeded in 24-well plates and incubated in a complete medium. After MSC(AT)s reached ~50% confluency, they were prepared for IF staining. HGFs were seeded in 24-well plates and divided into four groups (see Section 4.6, Cell migration assay). For cell IF staining, cells were washed with PBS three times and fixed with 4% paraformaldehyde for 15 min, after which they were treated with 0.1% Triton X-100 for 20 min, blocked by 5% normal goat serum for 30 min at RT, and incubated at 4 °C overnight with antibodies, COL1A1 (A16891, Abclonal, Wuhan, China), IL-1RA (ab124962, Abcam, Cambridge, UK), and FN (A16678, Abclonal, China). The next day, cells were rewarmed at room temperature (RT) for 30 min and incubated with secondary antibodies (ab96899, Abcam, Cambridge, UK or ZLI-0316, ZSGB-BIO, China) for 1 h. After being washed with PBS three times for 5 min each, they were stained with a fluorescent mounting medium with DAPI. Images were acquired with an Olympus microscope (Olympus Corporation, Tokyo, Japan).

### 4.8. Western Blot

Proteins from cells or exosomes were suspended in sodium dodecyl sulfate (SDS) loading buffer. After boiling, protein samples were separated on a 10% SDS-PAGE Gel and transferred to polyvinylidene fluoride membranes (Millipore, Darmstadt, Germany). Next, the membranes were incubated with primary antibodies against CD9 (ab263019, Abcam, Cambridge, UK), CD63 (ab134045, Abcam, Cambridge, UK), calnexin (ab133615, Abcam, Cambridge, UK), IL-1RA (ab124962, Abcam, Cambridge, UK), and β-Actin (TA-09, ZSGB-BIO, China) at 4 °C overnight, and then with secondary antibodies against rabbit (ZB-2301, ZSGB-BIO, China) or mouse (ZB-2305, ZSGB-BIO, China) for 1 h at RT. Finally, the signals were visualized. 

### 4.9. Formation and Characterization of a Hydrogel Loaded with Exosomes

The injectable hydrogel was prepared according to the manufacturer’s instructions (EFL-GM-30, Engineering For Life, Suzhou, China). Exosomes were mixed with hydrogel gently in equal volumes. To observe the distribution of exosomes in the hydrogel, exosomes were labeled with PKH26, and the images were taken using a confocal imaging system (TCS-SP8, Leica, Germany). To determine the release efficiency of exosomes in the hydrogel, the hydrogel-exosome complex was placed into a 48-well plate containing physiological saline. In order to reduce water evaporation, an equal volume of physiological saline was placed in the surrounding wells. The well plate was incubated in a constant temperature shaker at 37 °C with a speed of 60 rpm. Physiological saline of the soaked material was collected for the bicinchoninic acid assay (BCA assay). According to the protocols, the BCA Protein Assay Kit (23225, Thermo Fisher Scientific, Waltham, MA, USA) was used to detect the protein content in physiological saline.

### 4.10. qRT-PCR 

Total RNA was isolated from MSC(AT)s using TriZol Reagent (Invitrogen, Thermo Fisher Scientific, Waltham, MA, USA) and reverse-transcribed (RR036A, Takara, Osaka, Japan) into complementary DNA (cDNA). The produced cDNAs were used for the following experiments. qRT-PCR was performed to detect the gene expressions by the ABI Prism 7500. β-actin was used as the internal control, and the relative mRNA abundance of target genes was quantified with β-actin and calculated with the 2^−ΔΔCT^ method. Primer sequences for target genes are listed in Appendix A.

### 4.11. Animals

Next, 6-8-week-old male C57BL/6N mice were obtained from Beijing Vital River Laboratory Animal Technology Co., Ltd. (Beijing, China) Mice were maintained under specific pathogen-free conditions with a temperature of 22 ± 1 °C, relative humidity of 50 ± 1%, and a light/dark cycle of 12/12 h and were given standard chow and water. They were kept in a specific pathogen-free facility for one week to adapt to the environment. 

All animal studies (including the mice euthanasia procedure) were approved by the Ethics Committee of the Peking University Health Science Center (LA2018017). In addition, all animal experiments were performed according to the regulations and guidelines of the Peking University Institutional Animal Care and Use Committee and ARRIVE guidelines. 

### 4.12. Induction of an MRONJ-like Mouse Model and Exosome Treatment

Mice were randomly divided into four groups (*n* = 5/group): (1) Ctrl group: treated with a vehicle and PBS at the extraction site; (2) Zol group: treated with zoledronate and PBS at the extraction site; (3) Hydrogel group: treated with zoledronate and hydrogel at the extraction site; (4) Exo group: treated with zoledronate and hydrogel loaded with Exo at the extraction site. To establish an MRONJ model, mice were administered 0.9% NaCl saline (Veh) or zoledronate (1 mg/kg, SML0223, Sigma-Aldrich, St. Louis, MO, USA) intraperitoneally once a day for four weeks during the experiment process. After two weeks of drug treatment, the maxillary first molars of mice were extracted under general anesthesia using pentobarbital (50 mg/kg, P3761, Sigma-Aldrich, St. Louis, MO, USA). After tooth extraction, PBS was locally injected into the tooth sockets for the Ctrl group or the Zol group. For the Hydrogel group, the hydrogel was locally injected into the tooth sockets, while an Exo group received a local injection of a hydrogel loaded with Exo into the tooth sockets. A 405 nm light source was then used to irradiate the tooth sockets for 20 s to solidify the hydrogel after injection. 

In the subsequent rescue animal studies, mice were randomly divided into five groups (*n* = 5/group): (1) Ctrl group: treated with a vehicle and PBS at the extraction site; (2) Zol group: treated with zoledronate and PBS at the extraction site; (3) Hydrogel group: treated with zoledronate and hydrogel at the extraction site; (4) Exo (si-scramble) group: treated with zoledronate and hydrogel loaded with Exo (si-scramble) at the extraction site; (5) Exo (si-IL-1RA) group: treated with zoledronate and hydrogel loaded with Exo (si-IL-1RA) at the extraction site. The operation was carried out following the protocol mentioned above. Two weeks after extraction, mice were euthanized by cervical dislocation. The maxillae were harvested and fixed in 4% paraformaldehyde overnight and subjected to further analyses.

### 4.13. Patients 

Gingival samples were taken from 5 MRONJ patients and 5 healthy controls who underwent MRONJ-related or orthopedic surgeries at Peking University School and the Hospital of Stomatology. The collection methods, surgical approach, and inclusion and exclusion criteria are described in a previous study [44]. The harvested tissues were fixed in 4% paraformaldehyde for further studies. The Institutional Review Board (IRB) of Peking University Hospital of Stomatology approved the current study (PKUSSIRB-202170184). All enrolled individuals provided written informed consent. 

### 4.14. MicroCT Analysis

Micro-computed tomography (microCT) was used to evaluate new bone formation in the extraction sockets. The following microCT parameters were applied: 60 kV, 2 mA, J. Morita Corp., Kyoto, Japan). Three-dimensional image analysis software (Inveon Research Workplace (SIEMENS, Munich, Germany) was used to construct three-dimensional images and perform bone morphometric analysis of the extraction sockets. BV/TV, BMD, and Tb.Th were measured by an experimenter blind to experimental conditions. Image J 1.51j8 software was used to analyze the data.

### 4.15. Histology

Fixed maxillae were decalcified with 10% ethylenediaminetetraacetic acid (EDTA) at 37 °C for 2 weeks, embedded in paraffin, and sectioned into 4-μm-thick sections. H&E staining was used for histological observations, while Masson’s trichrome staining was applied to determine the degree of collagen maturity according to the instructions (G1340, Solarbio, China). TRAP staining was used to detect TRAP-positive cells per bone marrow area (#/mm^2^) to indicate osteoclasts according to the instructions (387A, Sigma-Aldrich, St. Louis, MO, USA). 

### 4.16. Tissue Immunochemistry Staining

In brief, sections (4 µm) were made and deparaffinized. After dewaxing the xylene, the sections were rehydrated with an alcohol gradient, and endogenous peroxidase was eliminated with 3% H_2_O_2_ in the dark for 20 min. Sections were then placed in 0.01 M sodium citrate buffer solution at 98 °C for 20 min, cooled at RT, and then mixed with anti-OCN antibodies (A6205, Abclonal, China), anti-IL-1β antibodies (ab9722, Abcam, Cambridge, UK), anti-TNF-α antibodies (ab1793, Abcam, Cambridge, UK), anti-CD86 antibodies (19589, CST, Boston, MA, USA), or ant-CD206 antibodies (24595, CST, Boston, MA, USA) at 4 °C overnight. Next, they were rewarmed at RT for 30 min and washed in PBS three times for 5 min each. Secondary antibodies (PV-9001, ZSGB-BIO or PV-9002, ZSGB-BIO, China) were incubated for 20 min and then stained with a DAB kit (ZLI-9018, ZSGB-BIO, China). Images were acquired with an Olympus microscope.

### 4.17. Tissue Immunofluorescent Staining 

For IF staining, sections were deparaffinized. After antigen blocking and retrieval, the sections were incubated with anti-IL-1RA antibodies (ab124962, Abcam, Cambridge, UK) at 4 °C overnight. They were rewarmed at RT for 30 min and thoroughly rinsed with PBS. Secondary Dylight 488 conjugated goat anti-rabbit antibodies (ab96899, Abcam, Cambridge, UK) were added and incubated at 37 °C for 1 h and stained with a fluorescent mounting medium with DAPI. A confocal imaging system (TCS-SP8, Leica, Germany) was used to analyze all images.

### 4.18. Statistical Analysis

All data are presented as the mean value-standard deviation. Unpaired two-tailed Student’s *t*-tests were performed to determine the differences between the two experimental groups. Analysis of variance (ANOVA) was performed to determine the differences among three or more experimental groups. Two-tailed *p* values < 0.05 (*), <0.01 (**), <0.001 (***), and <0.0001 (****) were considered significantly different.

## 5. Conclusions

Our results suggested that MSC(AT)-derived exosomes could effectively accelerate gingival wound healing and prevent MRONJ by IL-1RA-mediated pro-inflammation signaling suppression. This study might help elucidate the mechanisms of MRONJ and promote the early clinical prevention of MRONJ by MSC(AT)-based therapy. 

## Figures and Tables

**Figure 1 ijms-24-08694-f001:**
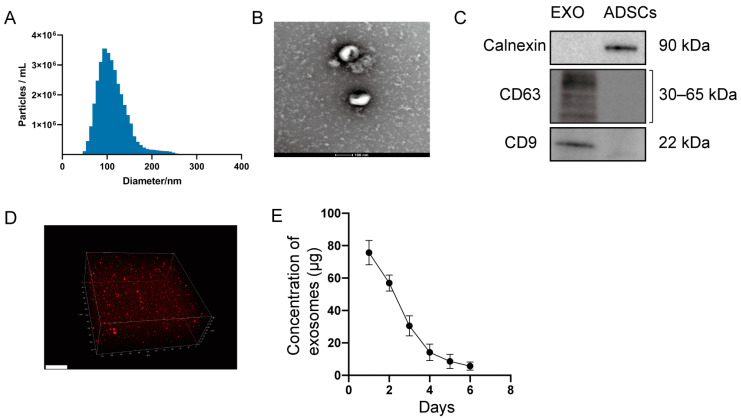
MSC(AT)-derived exosomes. (**A**) The particle size distribution of exosomes measured by NTA analysis. (**B**) Representative transmission electron micrographs of exosomes. Scale bar = 100 nm. (**C**) Western blot showing the expression of CD9, CD63, and calnexin in MSC(AT)s and MSC(AT)-derived exosomes. (**D**) Representative image of PKH26-labelled exosomes (red dots) encapsulated in the hydrogel with a confocal microscope. Scale bar = 50 μm. (**E**) The release of exosomes from hydrogel in vitro by BCA assay.

**Figure 2 ijms-24-08694-f002:**
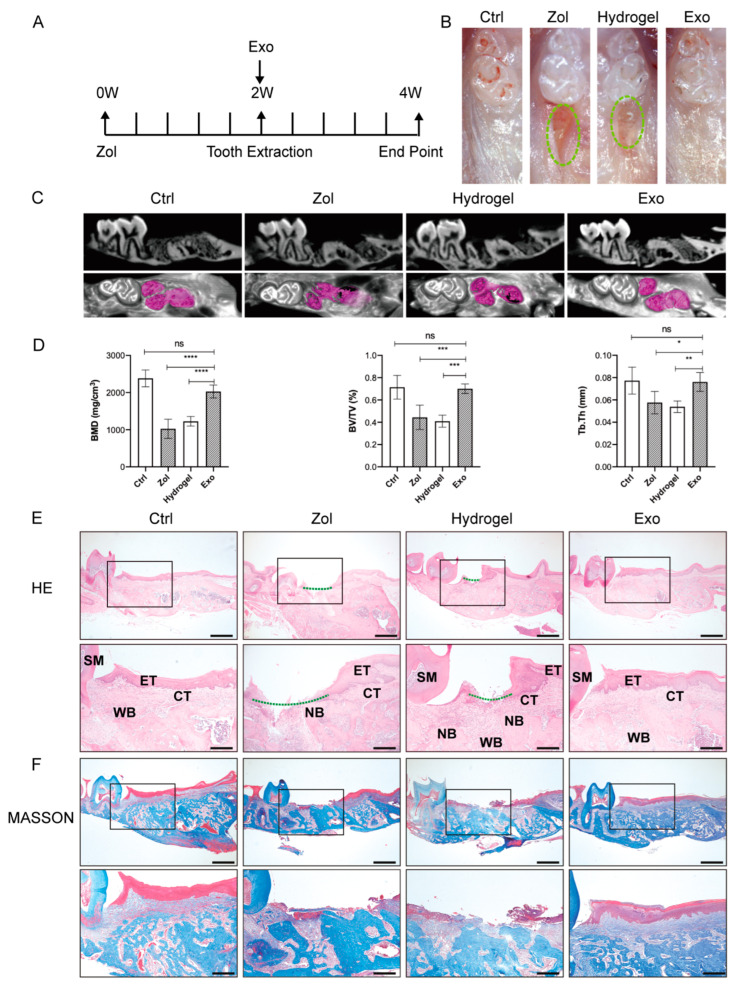
Exosomes accelerate primary gingival healing of tooth sockets in MRONJ-like mice. (**A**) A graphic representation of the timeline used in the study. (**B**) Clinical evaluation of the osteomucosal tissues at the tooth-extracted areas photographed 2 weeks after tooth extraction (Ctrl (non-drug treatment) group, Zol (treated with Zol) group, Hydrogel (treated with Zol and hydrogel) group, and Exo (treated with Zol and hydrogel loaded with exosomes) group). Green dot circles represent the unhealed tooth extraction area. (**C**) MicroCT image of tooth extraction sockets. The red area indicates the extracted parts. (**D**) Quantification of BMD, BV/TV, and Tb.Th in each group. (**E**) Evaluation of 2 weeks after tooth extraction by hematoxylin and eosin (H&E) staining. Black square: areas were magnified. The Green dotted line represents unhealed gingival mucosa. SM, second molar; ET, epithelial tissue; CT, connective tissue; NB, necrotic bone; WB, woven bone. Scale bar = 500 μm (upper), scale bar = 200 μm (lower). (**F**) Evaluation of 2 weeks after tooth extraction by Masson staining. Black square: areas were magnified. Scale bar = 500 μm (upper), scale bar = 200 μm (lower). (* *p* < 0.05, ** *p* < 0.01, *** *p* < 0.001, **** *p* < 0.0001, ns: not significant).

**Figure 3 ijms-24-08694-f003:**
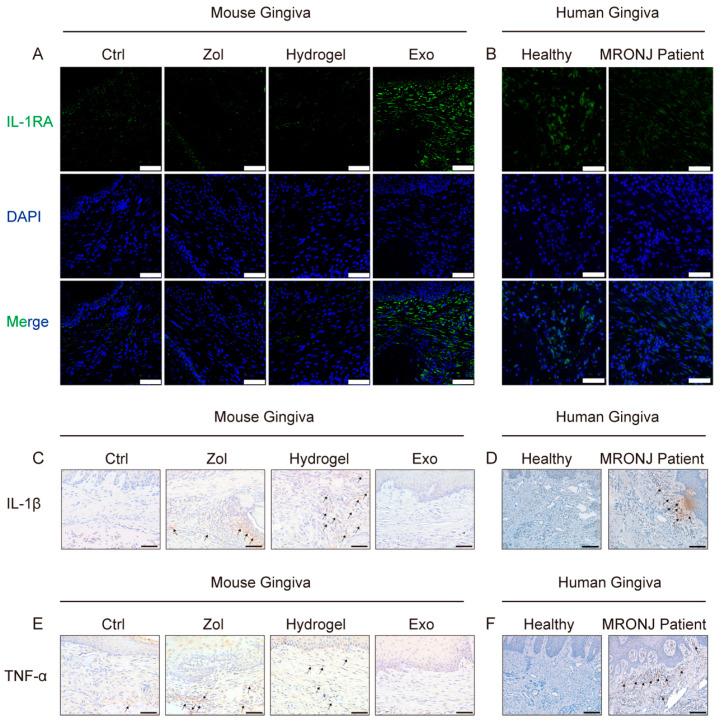
Exosomes increase IL-1RA expression and decrease TNF-α and IL-1β expression in the gingiva tissue. (**A**) Images of IL-1RA IF staining of mouse gingiva tissue in each group (Ctrl (non-drug treatment) group, Zol (treated with Zol) group, Hydrogel (treated with Zol and hydrogel) group, and Exo (treated with Zol and hydrogel loaded with exosomes) group). Scale bar = 50 μm. (**B**) Images of IL-1RA IF staining of human gingiva tissue in each group. Scale bar = 50 μm. (**C**) Images of IL-1β IHC staining of mouse gingiva tissue in each group. Black arrowhead: IL-1β-positive area. Scale bar = 50 μm. (**D**) Images of IL-1β IHC staining of human gingiva tissue in each group. Black arrowhead: IL-1β-positive area. Scale bar = 100 μm. (**E**) Images of TNF-α IHC staining of mouse gingiva tissue in each group. Black arrowhead: TNF-α-positive area. Scale bar = 50 μm. (**F**) Images of TNF-α IHC staining of human gingiva tissue in each group. Black arrowhead: TNF-α-positive area. Scale bar = 100 μm.

**Figure 4 ijms-24-08694-f004:**
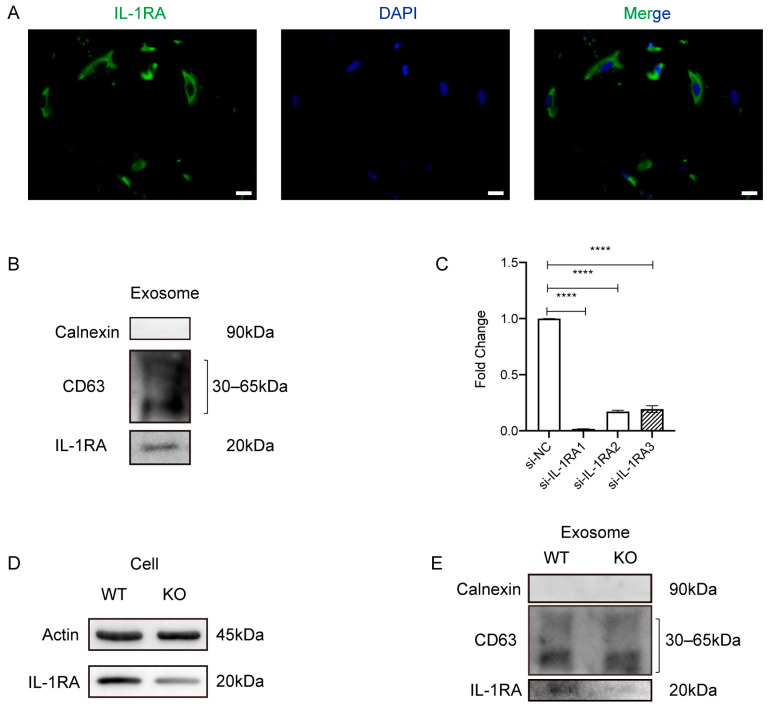
*IL-1RA* gene silencing. (**A**) Immunofluorescent staining of IL-1RA (green) in MSC(AT)s. Scale bar = 20 μm. (**B**) Protein levels of calnexin, CD63, and IL-1RA from exosomes isolated from MSC(AT)s assessed using Western blot. (**C**) *IL-1RA* gene silencing efficiency detected by mRNA expression. (**** *p* < 0.0001). (**D**) *IL-1RA* gene silencing efficiency detected by Western blot of MSC(AT)s. (**E**) *IL-1RA* gene silencing efficiency detected by Western blot of exosomes isolated from MSC(AT)s.

**Figure 5 ijms-24-08694-f005:**
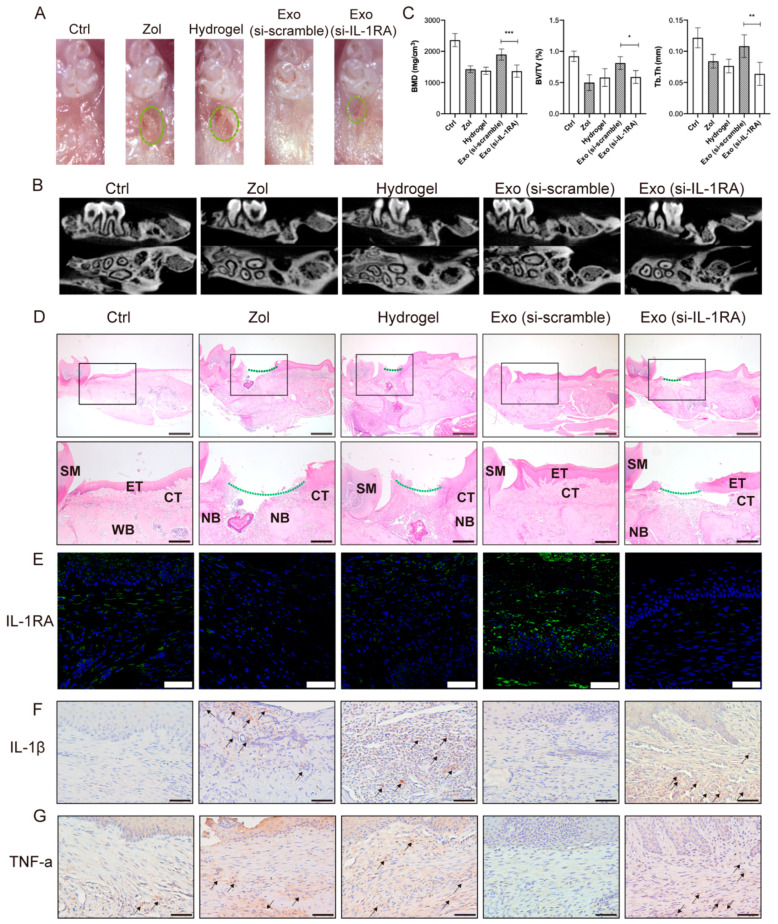
Knockdown of IL-1RA in MSC(AT)s attenuates the ability of exosomes to promote gingival closure and MRONJ healing. (**A**) Clinical evaluation of the osteomucosal tissues at the tooth-extracted areas, 2 weeks after tooth extraction (Ctrl (non-drug treatment) group, Zol (treated with Zol) group, Hydrogel (treated with Zol and hydrogel) group, Exo (si-scramble) (treated with Zol and hydrogel loaded with exosomes (si-scramble)) group), and Exo (si-IL-1RA) group (treated with Zol and hydrogel loaded with exosomes (si-IL-1RA)) group). Green dot circles: unhealed tooth extraction area. (**B**) MicroCT image of tooth extraction sockets. (**C**) Quantification of BMD, BV/TV, and Tb.Th in each group. (**D**) Evaluation 2 weeks after tooth extraction by H&E staining. Black square: areas were magnified. Green dotted line: unhealed gingival mucosa. SM, second molar; ET, epithelial tissue; CT, connective tissue; NB, necrotic bone; WB, woven bone. Scale bar = 500 μm (upper), scale bar = 200 μm (lower). (**E**) Images of IL-1RA IF staining of gingiva tissue in each group. (**F**) Images of IL-1β IHC staining of gingiva tissue in each group. Black arrowhead: IL-1β-positive area. Scale bar = 50 μm. (**G**) Images of TNF-α IHC staining of gingiva tissue in each group. Black arrowhead: TNF-α-positive area. Scale bar = 50 μm (* *p* < 0.05, ** *p* < 0.01, *** *p* < 0.001).

**Figure 6 ijms-24-08694-f006:**
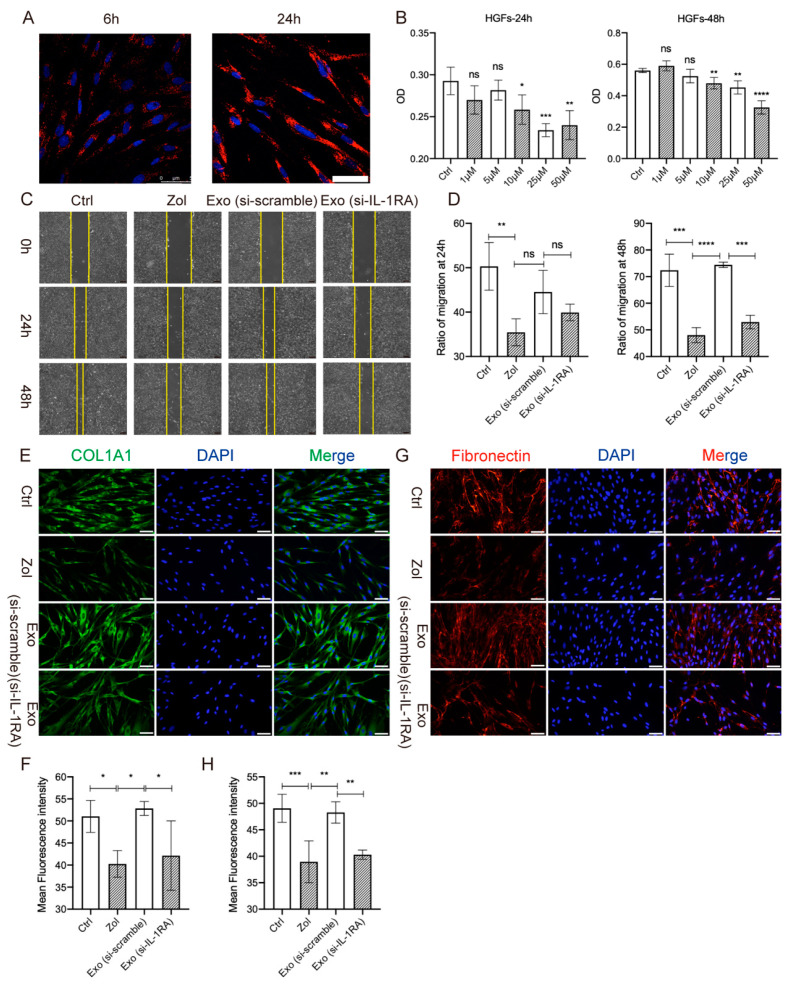
Knockdown of IL-1RA in MSC(AT)s attenuates MSC(AT)-derived exosomes and promotes HGF migration. (**A**) Uptake analysis of PKH26-labeled exosomes by HGFs. Scale bar = 20 μm. (**B**) CCK8 results showed the effects of Zol at different concentrations on HGF proliferation at 24 h and 48 h. (**C**) Wound-healing assay of HGFs treated with Zol and Exo (si-scramble) or Exo (si-IL-1RA) at 24 h and 48 h. Scale bar = 200 μm. Ctrl: blank control; Zol: cells incubated with Zol (10 μM); Exo (si-scramble): cells incubated with Zol (10 μM) and 50 μg/mL Exo (si-scramble); Exo (si-IL-1RA): cells incubated with Zol (10 μM) and 50 μg/mL Exo (si-IL-1RA). (**D**) Quantification of the wound-healing assay at 24 h and 48 h. (**E**) Images of COL1A1 IF staining in HGFs. Scale bar = 50 μm. (**G**) Images of FN IF staining in HGFs. Scale bar = 50 μm. (**F**,**H**) Quantifying the expression of COL1A1 and FN in each group. (* *p* < 0.05, ** *p* < 0.01, *** *p* < 0.001, **** *p* < 0.0001, ns: not significant).

## Data Availability

Data are contained within the article.

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
