# Peer review of "Exosomes Derived from Adipose Tissue-Derived Mesenchymal Stromal Cells Prevent Medication-Related Osteonecrosis of the Jaw through IL-1RA"

_ijms, 2023, doi:10.3390/ijms24108694_

Round 1

Reviewer 1 Report

The article entitled "Exosomes Derived from Adipose-Derived Stem Cells Prevent Medication-Related Osteonecrosis of the Jaw Through an IL- 3 1RA" written by Yi Zheng and colleagues evaluates the possible application of extracellular vesicles isolated from adipose derived stromal cells to prevent medication-related osteonecrosis of the jaw. 

The article is potentially of interest to the journal, but there are some conceptual problems underlying it that are very relevant and for which, while I am inclined to reject it, I would like clarification from the authors.

-Firstly, the authors speak of stem, but they refer to a population of mesenchymal stromal cells from adipose tissue. The word stem now in reference to mesenchymal cells is a terminology no longer in use as also defined by the ISCT itself.

-In the representation of the NTA profile, they should reduce the scale to allow a true appreciation of the size range of the EVs and also give information about the mean, fashion and SD.

-In WB as there is variability in the expression of typical EV markers (10.3390/cells10112948), the authors should show all three markers i.e. CD9, CD81 and CD63.

-In CTR the wound heals as well as EV treatment, which is an indication that the wound healing process of the tissue is working effectively. Treatment with zoleidronic acid should mimic the pathology, so EV treatment should be carried out (if a preventive effect is considered) at the same time as treatment with zoleidronic acid itself. Or do a pre-treatment with EVs and then expose to zoleidronic acid.

-In figure 4 the CD63 blot cannot be evaluated, the authors should propose another one and also the other markers as previously indicated.

Reviewer 2 Report

Dear colleagues!
After review of the manuscript by Zheng et al. I have the following comments as a Reviewer assigned by the Editor. Generally, it is an original and well-designed study that addresses an important unmet medical need treatment. Most appropriate experiments to support the conclusions are present and performed in a sound manner. The text is well-proofed and has sufficient quality with minimal editing potentially required once accepted. Nevertheless, the Reader may pose several points:

1) loading controls (e.g. tubulin, GAPDH) are missing in WB panels

2) has exosome release/retain from used hydrogel been assessed?

3) HGF migration is an important functional outcome yet it would be of interest whether myofibroblast differentiation is modulated? Role of MFB is wound closure and col I/FN deposition is critical and this process may be an important link to connect the therapy and outcome. Detection  MFB (derived from HGFs by TGFb induction) by a-SMA staining in vitro after Exo treatment (including IL-1RA pull-down) could be of interest.

4) TNFa and IL-1b have been detected in the tissue yet no impact on immune cells invasion/infiltration is presented in histological sections

5) Does IL-1RA pulldown or treatment by IL-1RA depleted exosomes influence ADSC differentiation?

Round 2

Reviewer 1 Report

Thank you the authors for answer my question

Author Response

Thank you for your comment.

Reviewer 2 Report

Dear colleagues!

I feel that Points 1-3 in my initial review have been properly addressed and can be dropped.

Yet for â„–4 and 5 I do suggest additional experimental procedures to clarify on mechanisms that are involved in described findings to strengthen the Manuscript and comply to growing reputation of the Journal.

Regards, Reviewer.
